# Optimal Financing Strategy in a Capital-Constrained Supply Chain with Retailer Green Marketing Efforts

**Xiaoli Zhang** [1,2], **Guoyi Xiu** [1], **Fakhar Shahzad** [3,*] and **Yupeng Duan** [4]

1   School of Economics and Management, Harbin University of Science and Technology, Harbin 150080, China; zhangxl@hrbfu.edu.cn (X.Z.); xiuguoyi@hrbust.edu.cn (G.X.)
2   Department of Accounting, Harbin Finance University, Harbin 150030, China
3   School of Management, Jiangsu University, Zhenjiang 212013, China
4   Department of Statistics and Applied Probability, University of California, Santa Barbara, CA 93106, USA; zhdhc@163.com
*   Correspondence: fshahzad51@yahoo.com

**Abstract:** The purpose of this research is to examine the green supply chain (GSC) financing decisions of manufacturers and capital-constrained retailers in order to establish a Stackelberg game model under decentralized and centralized decision-making. This paper studies the influence of retailers' choice of trade credit or bank loan financing strategy on a GSC's performance and analyzes their decision-making tendency. The results show that manufacturers should provide trade credit and participate in retailers' financing decisions to avoid double marginal effects under both centralized and decentralized decision-making. Interestingly, the optimal value of green marketing effort and retailer order quantity was twice as high as the decentralized under the centralized decision, indicating that the centralized decision could better improve GSC's financing efficiency. Especially when the trade credit financing strategy is feasible, this effect is more significant. Finally, the outcomes are verified through numerical simulation, which references GSC practitioners in management decisions.

**Keywords:** green supply chain; capital constraint; trade credit; bank loan; financing strategy

## 1. Introduction

The massive impact of global warming is increasing, and it affects economies globally. In the current era, with sustainable development strategies, excessive resource consumption, and ecological environment imbalance continue to ferment [1], and people begin to realize the harm caused by the mismatch between production and consumption to the environment. The enactment of environmental policies, consumers' green expectations, and the strengthening of environmental protection concepts have led them to change their shopping lists and prefer green products [2]. Due to the lack of the existing literature, a generally accepted definition of a green product has not been entirely determined. However, generally, green products are called "ecological products" or "environmentally friendly products" [3]. Moreover, Sdrolia and Zarotiadis [4] concluded that "green is a product (tangible or intangible) that minimizes its environmental impact (direct and indirect) during its whole life cycle, subject to the present technological and scientific status." Therefore, environmental sustainability, product greening, and other issues have attracted extensive attention from relevant government departments and green supply chain management (GSCM) researchers [5–7]. In addition, most organizations have started to use green product development and green marketing strategies to protect the environment and satisfy consumer preferences to obtain long-term profits [8].

Green marketing originates from developed countries' environmental protection regulations and policies in Europe and America for various industries. At present, it has been confirmed that green marketing plays a crucial role in promoting green products to achieve green supply chain (GSC) sustainability [9]. From the perspective of environmental

protection, green marketing refers to the marketing efforts made in the design, production, packaging, and promotion of products or services, including product planning, green advertising, green pricing, and ecological labeling [10]. On the one hand, green marketing can promote clean production through the development of green products, and on the other hand, it can promote sustainable consumption [11]. For example, Walmart was the first retailer to set an 18% reduction target for 2025, in line with the 2015 Paris Agreement, under which it will combine energy efficiency measures with 50% renewable energy. It also promises to work with suppliers to reduce waste on goods and packaging, protect natural resources, and achieve 100% recyclable packaging for private brands by 2025 [12]. In addition, new technologies were adopted to transform existing stores and create green energy efficiency marts. This green marketing method can turn consumers' green consciousness into actual purchasing action [13]. Consumers determine the environmental attributes of a product by its greenness. According to a global survey conducted by Accenture, more than 80% of respondents consider the product's greenness when purchasing decisions [14]. In the market, consumers' strong demand for green products will change the sales environment. Therefore, companies need to adopt green marketing strategies to compete for green opportunities [15].

However, in the two-stage GSC, manufacturers are responsible for the development and production of green products to meet sustainable green consumption and encourage green marketing of retailers. Which is also a manifestation of enterprises' fulfillment of social responsibility and a competitive advantage in establishing corporate image in the market [8]. At the same time, retailers can adopt environmental advertisements or environmental education to promote green products. The current popular green marketing strategy of live broadcast sales has attracted the attention of many consumers. Such media publicity and retailers' online and offline channel displays are the main ways for most consumers to understand, judge, and finally choose green products, which we call green marketing efforts [16]. Meanwhile, effective green marketing efforts can help consumers to understand the meaning of green products and experience the effects of green products [17]. Green marketing efforts can also enhance the consumers' concept of green environmental protection, stimulate the demand for green products, expand market demand, and increase the market share [16]. Considering this important issue and the lack of theoretical knowledge, it is feasible and necessary to consider retailers' green marketing efforts, especially in the capital-constrained supply chain.

Because of the sudden coronavirus (COVID-19), the GSC operation is faced with the risk of capital interruption and the shortage of capital that makes the participants unable to implement the optimal operation/marketing decision. Thus, one of the most important decisions faced by capital constraint retailers is financing their operations [5]. To solve this problem, retailers could use bank loans or trade credits to finance their everyday operations. This is an effective marketing effort, which can significantly improve product awareness and market demand [18]. In the GSC, the marketing-process greenness can increase retailers' sales revenue and promote its sustainable development. However, retailers need to pay extra efforts in green marketing during the sales process, and their marketing costs will increase accordingly, which is undoubtedly a further blow to the retailers under the constraint of capital. Therefore, it is practical to study how to implement green marketing efforts in the GSC and solve retailers' financing decision-making issues under the capital constraint, which has not been studied previously.

Most of the existing research results independently study the financing mode selection, carbon emission reduction, and green investment of SC subjects. While this paper also considers the level of green marketing efforts and financing decision-making process of retailers under the capital constraint. To fill the existing gap, this study is focusing on answering the following questions:

1. What is the optimal retail price and the level of green marketing effort when the retailer adopts different financing strategies under decentralized and centralized decision-making?

2.  What are the effects of green sensitivity and financing rates on the level of green marketing efforts and SC participants' profits?
3.  Under decentralized and centralized decision-making, how should the retailers with capital constraints choose the best financing strategy?

In order to answer these questions, this paper selects a GSC composed of a manufacturer and a capital-constrained retailer as the research object and establishes a Stackelberg game model solved by the inverse inference method. The Stackelberg game involves players with asymmetric functions, that is, leaders and followers. The leader decides the strategy first, and the followers choose the policy after understanding the leader's strategy. The leader predicts the best response of the follower and decides the most appropriate point [19]. Meanwhile, the Stackelberg game model is proposed, and used to find the balance point of maximizing the profit of each member of the supply chain [20]. In the GSC, the change rules of the key decision variables such as the optimal wholesale price, the optimal retail price, and the optimal green marketing effort level (when the capital-constrained retailers adopt the two financing modes such as trade credit and bank loan) are analyzed. Therefore, we adopted the Stackelberg game model, which is often used to study such dynamic issues as employed and recommended by [20–24]. Moreover, SC's profit distribution was also analyzed in this study to fill the existing research gap.

## 2. Literature Review of Key Points

### 2.1. Traditional SC Operation and Financing

Early researchers paid attention to the influence of capital constraint on the operational decision of the traditional SC. Goyal first used the Economic Order Quantity (EOQ) model to discuss the optimal order quantity decision under the condition that the supplier allows the retailer with the capital constraint to delay payment [25]. Huang extended the model to study the situation where suppliers allow retailers to make partially deferred payments. The retailer's inventory system is modeled as a cost minimization problem to determine its optimal inventory cycle and optimal order quantity [26]. From the perspective of game theory, Buzacott and Zhang [27] considered asset financing in production and operation, studied corporate asset mortgage considered the problem of asset financing in production and operation from the perspective of game theory, studied the issues of enterprises' asset-backed financing and joint production decision-making, and found that compared with their own funds, enterprises prefer bank loan financing. Xu and Birge [28] studied how capital constraints affected enterprises' production decisions and showed a correlation between enterprises' production and financing decisions. Similarly, Chao, Chen, and Wang [29] studied the problem of multi-period inventory control with capital constraints, which represented the enterprise's optimal operating policy's laziness on its capital status. Moreover, it is also confirmed the importance of incorporating financial constraints into corporate operating decisions [30]. The study of Seifert et al. [31] comprehensively elaborated on the motives of trade credit from management's perspective.

In addition, several scholars have explored the coordination issues under SC financing and found that trade credit is a beneficial tool for SC coordination, improving SC efficiency, and ensuring its sustainability [21,32–35]. The above scholars have proved the relationship between SC operation and financing from game theory and management. However, the financing method was single, and there is no comparative choice of SC financing strategies. Meanwhile, since the last decade, scholars have gradually broadened the research field, carried out in-depth research comparing internal and external financing models in the traditional SC, and extended financing channels. This will help improve the SC's efficiency, reduce financing costs, and effectively solve expensive and challenging financing for small and medium enterprises (SMEs). SC's internal financing refers to the manufacturer's trade credit to its downstream retailers, while the SC's external financing mainly refers to loans from financial institutions such as banks. Kouvelis and Zhao [36] constructed a Stackelberg model with suppliers as the leader and pointed out that the optimal trade credit contract

can maximize suppliers' profits and the entire SC. At the same time, retailers also prefer trade credit.

### 2.2. SC with Green Marketing

The term "green marketing," was first proposed in 1975 at a seminar organized by the American Marketing Association, also known as ecological marketing, environmental marketing, or sustainable marketing [37]. It refers to all consumers' actions and includes a wide range of marketing activities (such as pricing, planning, processing, production, and promotion). The purpose of these marketing activities is to minimize the negative impact on the environment in pursuing corporate goals [38]. Sustainable development is the theme of future green marketing that leads to societal change [39]. Different SC participants can implement different sustainable development strategies. For example, manufacturers can produce green products by reducing carbon emissions and green innovation, while retailers can implement environmental protection through green marketing efforts such as advertising and promotion [16]. As a common way to encourage sustainable behaviors, green marketing has been included in the theory of SC sustainable development [40]. Considering the green marketing behavior of retailers, this paper can fill this gap and broaden GSC's research.

With the increasing awareness of environmental protection, consumers are willing to choose green products as an effective way to protect the environment [2,41]. The study of Ko et al. [42] posits green marketing in products and sales from the perspective of consumers. Research shows that green marketing behaviors greatly affect corporate social responsibility and shape corporate image, thereby increasing its market share [8]. This consumer preference will force retailers to adopt green marketing and promote corporate social responsibility. Phan et al. confirmed that market demand is affected by retailers' green marketing efforts [43]. Mondal and Giri [44] constructed a two-period green closed-loop SC model composed of a single manufacturer and a single retailer and studied the impact of green innovation, marketing efforts, and the recycling rate of waste products on SC decisions. It is pointed out that manufacturers and retailers can increase market demand by improving green innovation and increasing marketing efforts. Sana [45] builds a green product marketing model based on corporate social responsibility, which explores the pricing strategy of green marketing and non-green marketing competition. Green marketing has a positive impact on corporate image and business performance. For a company that wants to maintain a competitive advantage, especially in the retail sector, green marketing is mostly in its interest [46]. Therefore, measuring the impact of green marketing on the GSC is worthy of attention and research. As far as we know, this is the first attempt to use green marketing as a financing mechanism to promote GSC sustainability and to explore the role of green marketing in SC financing.

### 2.3. GSC Financing with Capital Constraint

With the development of ecological economics, GSC financing under financial constraints has attracted great attention from academia and industry. At present, scholars mainly promote the sustainability of the SC and improve the performance from the perspectives of carbon emission reduction and green investment. Aljazzar et al. [47] organized the SC from the perspective of carbon emission cost and found that trade credit could improve the SC's environment and business performance. Considering the impact of carbon emission reduction on GSC operation and financing decisions, Dash Wu et al. [48] discussed the optimal order quantity of retailers, the optimal wholesale price of manufacturers, the optimal carbon emission level of bank loans, and trade credit financing. They found that the effect of trade credit financing was better than that of bank loans. Cao and Yu [49] studied the impact of carbon emissions on SC financing and performance and found that bank credit could alleviate the problem of overproduction and achieve optimal carbon emission reduction.

Another scholar studies the financing of the GSC from the perspective of green investment. They studied the optimal retail price and purchasing decision of retailers, the wholesale price of manufacturers, and green level decisions of products under the two-stage SC framework [50]. Yang et al. [5] considered a two-level GSC composed of a manufacturer and retailers with capital constraints and competition, designed revenue-sharing contracts from the SC's internal and external financing channels to coordinate the profit distribution GSC. Zhang [51] compared the optimal green decisions with and without green investments under manufacturers' capital constraints. Moreover, Fang and Xu [52] compared green credit financing and hybrid financing modes and concluded that manufacturers with capital constraints were also willing to make green investments.

In conclusion, scholars on trade credit and bank loans and other credit policy do much research; however, most research aimed at traditional SC financing and operations. Some studies focused on GSCM under the capital constraint without considering green marketing efforts [5,48,52,53]. Meanwhile, some other studies discussed green marketing efforts without paying attention to retailers' capital constraints [9,16]. This paper's research contents are compared with the existing research results, as shown in Table 1. Focusing on solving the retailer's capital constraints, integrating retailers' green marketing efforts into GSCM will help promote the coordinated development of the economy and the environment, and achieve the GSC's sustainability.

**Table 1.** Research contents of this paper in comparison with other related published studies.

| Literature | Capital Constraint | SC Financing Model | | | GSC Finance | | |
|---|---|---|---|---|---|---|---|
| | | Bank Financing | Trade Credit | Mixed Financing | Carbon Reduction | Green Investment | Green Marketing |
| Polonsky, 2011, Groening et al., 2018, Q. Zhu & Sarkis, 2016, Yenipazarli & Vakharia, 2015, Juvan & Dolnicar, 2017, Ko et al., 2013, Phan et al., 2019, Chang et al., 2019, Sana, 2020, Mukonza, 2020, Cui et al., 2020 | | | | | | | yes |
| Mondal & Giri, 2020 | | | | | | yes | yes |
| Goyal, 1985, Chao et al., 2008, Bai & Zhang, 2012, Seifert et al., 2013, C. H. Lee & Ä, 2010, C. H. Lee & Rhee, 2011, Du et al., 2013, Yan et al., 2016, Deng et al., 2018 | yes | | yes | | | | |
| Buzacott & Zhang, 2004, Ding & Wan, 2020 | yes | yes | | | | | |
| Feng et al., 2015, Kouvelis & Zhao, 2012, Jing et al., 2012, X. Chen, 2015, Yan & Sun, 2013, | yes | yes | yes | | | | |
| D. Chen et al., 2020, Honglin Yang et al., 2017 | yes | | | yes | | | |
| Aljazzar et al., 2018, | yes | | yes | | yes | | |
| Dash Wu et al., 2019 | yes | | yes | yes | yes | | |
| Cao & Yu, 2019 | yes | | yes | | yes | | |
| Haoxiong Yang et al., 2019, Luo et al., 2020 | yes | | yes | yes | | yes | |
| Zhang, 2020 | yes | | | | | yes | |
| Fang & Xu, 2020 | yes | | yes | yes | | yes | |
| Current Manuscript | yes | yes | yes | | | | yes |

## 3. Basic Model and Assumptions

### 3.1. Conceptual Definitions and Assumptions

This study considers a two-level GSC, which consists of a manufacturer as a leader and a retailer as a follower. We assume that the responsibility for green marketing efforts lies with the retailer, and the consumers in the marketplace are environmentally conscious,

and the retailer's green marketing efforts will affect market demand; based on [5,16,50] research work we establish the market demand function as;

$$q = \alpha - bp + \beta e, \tag{1}$$

where $q$ represents market demand, and the order quantity of the retailer equals the market demand, $q > 0$; $\alpha$ represents the potential size of the total market demand, $\alpha > 0$; $b$ represents the sales price sensitivity coefficient, $b > 0$; $p$ represents the retail price of the products; $\beta$ represents the market demand sensitivity to green marketing efforts, referred to as green sensitivity [16,54,55]. $e$ indicates the level of green marketing efforts, $0 \leq e \leq 1$; other notations' definition, and assumed variables are shown in Table 2.

**Table 2.** Notations definition.

| Parameters | Notations Definition |
|---|---|
| $\omega$ | Unit wholesale price of the products |
| $c_m$ | The manufacturer's unit production cost, and $p > \omega(1 + r_n) > c_m(1 + r_n)$. Resembles the assumptions of [36,53] |
| $r_n$ | $n = \{m, b\}$ Indicates the interest rate of the bank loan and trade credit respectively. |
| $g(e)$ | Green marketing cost [16], $g(e) = \frac{1}{2}se^2$, where $s$ represents the cost parameter for green marketing efforts, $s$ is a constant. |
| $\pi_j^i$ | The profits of GSC member enterprises, $i = \{N, M, B\}, j = \{m, r, t\}$ represent the profits of the manufacturer, the retailer and the overall SC under the strategies of no financing demand, trade credit financing, and bank loan financing. |

To ensure the concavity of the SC function of corporate profits, all related parameters are feasible without loss of generality [55], we need that $s$ is typically large [56]. In addition, the model also satisfies the following basic assumptions:

1. The SC members' operating behaviors follow the financing norms, assuming that both parties do not breach the contract [57].
2. All parties in the SC should try to neutralize the risk and share information to pursue profit maximization [21,36].
3. The manufacturer's production capacity can fully meet the retailer's market demand and aftermarket development; the product sales price remains unchanged [53].
4. Perfect market competition and the risk-free interest rate is 0 [36].

### 3.2. The Optimal Equilibrium Solution of the GSC without Capital Constraints

Firstly, this paper builds a benchmark model without capital constraints for retailers. In the benchmark model, only the influence of green sensitivity on equilibrium results is considered. Then, we design the financing models of trade credit and bank loan under centralized and decentralized decisions, respectively, and consider the influence of trade credit interest rate and bank loan interest rate on equilibrium results.

If the retailer has sufficient funds, its own funds can meet the expenditure of daily operations and green marketing efforts, and no financing is needed. It is helpful for manufacturers and retailers to choose the best financing strategy by analyzing different financing strategies. At this time, we describe the profit function of the manufacturer, retailer, and the whole supply chain without capital constraint. In this two-level GSC system, the manufacturer is accountable for producing green products and determines the wholesale price; the retailer is responsible for implementing green marketing behaviors in the market to promote green products, determines the sales price of green products $p$, and

the level of green marketing efforts $e$. At this point, the profits function of the manufacturer, the retailer, and the overall SC can be described as;

$$\pi_m^N(\omega) = (\omega - c_m)(\alpha - bp + \beta e), \tag{2}$$

$$\pi_r^N(p, e) = (p - \omega)(\alpha - bp + \beta e) - \frac{1}{2}se^2, \tag{3}$$

According to the Stackelberg game model sequence of the above two-stage GSC, The Leader, manufacturer, and The Follower, the retailer, adopts the reverse recursion method to obtain the optimal equilibrium result of the wholesale price without capital constraints:

$$\omega^{N^*} = \frac{\alpha + bc_m}{2b}, \tag{4}$$

Moreover, we can obtain the optimal equilibrium results of the retail price, the level of green marketing efforts, and the order quantity:

$$p^{N^*} = \frac{2\alpha bs + (bs - \beta^2)(\alpha + bc_m)}{2b(2bs - \beta^2)}, \tag{5}$$

$$e^{N^*} = \frac{\beta(\alpha - bc_m)}{2(2bs - \beta^2)}, \tag{6}$$

$$q^{N^*} = \frac{bs(\alpha - bc_m)}{2(2bs - \beta^2)}, \tag{7}$$

Finally, we can obtain the optimal profits of the manufacturer, the retailer, and the overall SC:

$$\pi_m^{N^*} = \frac{s(\alpha - bc_m)^2}{4(2bs - \beta^2)}, \tag{8}$$

$$\pi_r^{N^*} = \frac{s(\alpha - bc_m)^2}{8(2bs - \beta^2)}, \tag{9}$$

$$\pi_t^{N^*} = \frac{3s(\alpha - bc_m)^2}{8(2bs - \beta^2)}, \tag{10}$$

## 4. GSC Financing Strategies Under Decentralized Decision-Making

Green marketing efforts of retailers requires additional funding. The sudden epidemics of Coronavirus (COVID-19) increased the likelihood of a financial collapse of the SC. If the initial funds of retailers only support green marketing expenditure, it is challenging to guarantee regular operating capital needs. Financing is therefore essential, and the internal financing of the SCs, called trade credit financing, can be carried out by upstream SC manufacturers; external financing of SCs such as the financing of bank loans can also be chosen.

### 4.1. Trade Credit Financing Strategy

In the mode of trade credit financing, the interest rate of trade credit $r_m$ is an exogenous variable. The manufacturer first sets the wholesale price $\omega$, and the retailer determines the sales price $p$ and level of green marketing efforts $e$ keeping the wholesale price provided by the manufacturer in view. Throughout the business cycle, the manufacturer's profit is expressed as wholesale products' revenue plus loan interest, minus production costs. On the other hand, the retailer's profit includes the income from the sale of products minus the purchase cost, the interest in loan repayment, and green marketing. The corresponding profit is expressed as follows:

$$\pi_m^M(\omega) = (\omega - c_m)q + r_m L_M = (\omega(1 + r_m) - c_m)(\alpha - bp + \beta e), \tag{11}$$

$$\pi_r^M(p,e) = (p - \omega)q - r_m L_M - \frac{1}{2}se^2 = (p - \omega(1 + r_m))(\alpha - bp + \beta e) - \frac{1}{2}se^2 \quad (12)$$

where: $L_M = \omega q$.

As per the Stackelberg game model sequence of the above two-stage GSC, similar to the solution method in Section 3.2, the following proposition can be obtained:

**Proposition 1.** *Under the trade credit financing strategy, the optimal equilibrium results of the wholesale price, the retail price, the level of green marketing efforts, and the order quantity can be obtained as follows:*

$$\omega^{M^*} = \frac{\alpha + bc_m}{2b(1 + r_m)}, \quad (13)$$

$$p^{M^*} = \frac{\alpha(3bs - \beta^2) + b(bs - \beta^2)c_m}{2b(2bs - \beta^2)}, \quad (14)$$

$$e^{M^*} = \frac{\beta(\alpha - bc_m)}{2(2bs - \beta^2)}, \quad (15)$$

$$q^{M^*} = \frac{bs(\alpha - bc_m)}{2(2bs - \beta^2)}, \quad (16)$$

Moreover, we can obtain the optimal profits of the manufacturer, the retailer, and the overall SC under the trade credit financing strategy:

$$\pi_m^{M^*} = \frac{s(\alpha - bc_m)^2}{4(2bs - \beta^2)}, \quad (17)$$

$$\pi_r^{M^*} = \frac{s(\alpha - bc_m)^2}{8(2bs - \beta^2)}, \quad (18)$$

$$\pi_t^{M^*} = \frac{3s(\alpha - bc_m)^2}{8(2bs - \beta^2)}, \quad (19)$$

By comparing the relationship of $\omega^{M^*}$, $p^{M^*}$, $e^{M^*}$, $q^{M^*}$ and $\omega^{N^*}$, $p^{N^*}$, $e^{N^*}$, $q^{N^*}$, when compared with no capital constraints, it is found that, $\omega^{M^*} < \omega^{N^*}$, $p^{M^*} = p^{N^*}$, $e^{M^*} = e^{N^*}$, $q^{M^*} = q^{N^*}$.

When the manufacturer provides the trade credit financing, to tempt the retailer to maximize the order quantity and exploit the profits, the manufacturer must lower the wholesale price because you can also get loan interest income. The retail price, the level of green marketing efforts, and order quantity are not affected by capital constraints, and the decision is the same as without capital constraints. Simultaneously, consumers' environmental awareness and pursuit of green products will not be affected by the retailer's financial constraints. Proposition 1 proposes that the manufacturer's wholesale price will decrease as the interest rate $r_m$ increases, but the retailer's sales price will not change accordingly, mainly because the retailer needs more profits to repay the interest as scheduled.

*4.2. Bank Loan Financing Strategy*

Under the bank loan financing strategy, the bank provides the interest rate $r_b$. As the leader, the manufacturer first sets the wholesale price, and the retailer determines the retail price $p$ and the level of green marketing efforts $e$ according to the manufacturer's wholesale price. Throughout the business cycle, the manufacturer's profit only includes the revenue of wholesale products, and the retailer's profit includes the revenue of selling

products minus the cost of purchasing, repayment of loan interest, and green marketing costs. The corresponding profits can be expressed as follows:

$$\pi_m^B(\omega) = (\omega - c_m)q, \tag{20}$$

$$\pi_r^B(p,e) = (p - \omega)q - r_b L_B - \frac{1}{2}se^2, \tag{21}$$

where: $L_B = \omega q$.

Similarly, using the reverse recursion method to solve Formulas (20) and (21), the following propositions can be obtained:

**Proposition 2.** *Under the bank loan financing strategy, the optimal equilibrium results of the wholesale price, the retail price, the level of green marketing efforts, and the order quantity can be obtained as follows:*

$$\omega^{B^*} = \frac{\alpha + bc_m(1 + r_b)}{2b(1 + r_b)}, \tag{22}$$

$$p^{B^*} = \frac{\alpha(3bs - \beta^2) + b(1 + r_b)(bs - \beta^2)c_m}{2b(2bs - \beta^2)}, \tag{23}$$

$$e^{B^*} = \frac{\beta(\alpha - bc_m(1 + r_b))}{2(2bs - \beta^2)}, \tag{24}$$

$$q^{B^*} = \frac{bs(\alpha - bc_m(1 + r_b))}{2(2bs - \beta^2)}, \tag{25}$$

Moreover, we can obtain the optimal profits of the manufacturer, the retailer, and the overall SC under the bank loan financing strategy:

$$\pi_m^{B^*} = \frac{s(\alpha - bc_m(1 + r_b))^2}{4(1 + r_b)(2bs - \beta^2)}, \tag{26}$$

$$\pi_r^{B^*} = \frac{s(\alpha - bc_m(1 + r_b))^2}{8(2bs - \beta^2)}, \tag{27}$$

$$\pi_t^{B^*} = \frac{s(3 + r_b)(\alpha - bc_m(1 + r_b))^2}{8(1 + r_b)(2bs - \beta^2)}, \tag{28}$$

From Proposition 2, we can see that under the optimal decision of bank loan financing, the relationship between the wholesale price, the retail price, the level of green marketing efforts and the order quantity, and the interest rate of the bank loan is as follows:

$$\frac{\partial \omega^{B^*}}{\partial r_b} < 0, \ \frac{\partial p^{B^*}}{\partial r_b} > 0, \ \frac{\partial e^{B^*}}{\partial r_b} < 0, \ \frac{\partial q^{B^*}}{\partial r_b} < 0.$$

The increase in bank loan interest rate will cause retailers to repay the bank more interest, increase financing costs, and cause losses to their own profits. Retailers will persistently avoid it, and it will even reduce their motivation for green marketing efforts and order quantity. To encourage retailers to place orders and the manufacturer to reduce the wholesale prices, by comparing $\omega^{N^*}$ and $\omega^{B^*}$, we find that $\omega^{N^*} - \omega^{B^*} = \frac{\alpha r_b}{2b(1+r_b)} > 0$, when compared with no capital constraints, the wholesale price of the manufacturer has fallen.

**Proposition 3.** *Regardless of the strategies of financing, the wholesale price of the manufacturer, the sales price, the level of green marketing efforts, the order quantity of the retailer, and the green sensitivity all have the following relationship:*

$$\frac{\partial \omega^{X^*}}{\partial \beta} = 0, \ \frac{\partial p^{X^*}}{\partial \beta} > 0, \ \frac{\partial e^{X^*}}{\partial \beta} > 0, \ \frac{\partial q^{X^*}}{\partial \beta} > 0, \ (X = M, B).$$

Proposition 3 shows that the manufacturer's wholesale price remains unaffected by green sensitivity, while the retail price, the level of green marketing efforts, and the order quantity of the retailer will increase with the increase in the green sensitivity.

The greater the green sensitivity leads to the market's demand response effect, which will encourage retailers to exert more green marketing efforts, thereby increasing product ordering. The studies of [8,43,44] proposed the same in their research work. Manufacturers will consider increasing the wholesale price for their profit maximization. Simultaneously, retailers will also increase their sales price, and manufacturers' and retailers' profits will increase respectively and create a mutually beneficial SC situation.

*4.3. GSC Financing Strategy Selection*

The above sections explained that the optimal decision variables are affected by interest rates and green sensitivity. From the retailer's perspective, subject to capital constraints, considering how to choose two financing strategies, focus on this section.

**Proposition 4.** *When the trade credit and bank loan interest rates are the same, that is, $r_m = r_b$, comparing the most optimal choices among the two financing strategies, then $\omega^{B^*} \geq \omega^{M^*}$, $p^{B^*} \geq p^{M^*}$, $e^{B^*} \leq e^{M^*}$, $q^{B^*} \leq q^{M^*}$.*

Under the bank loan financing strategy, retailers need to repay bank interest, which weakens the motivation for green marketing efforts and harms consumers' interests, and increases product sales prices. It can be seen from the demand function that market demand is better under the trade credit financing strategy, which is more conducive to retailers to develop the market. Moreover, under this strategy, manufacturers will set lower wholesale prices to encourage retailers to place orders. How should retailers, who are in a disadvantaged position in the SC, participate in financing?

**Corollary 1.** *If $r_m = r_b$ both manufacturers with sufficient funds and retailers in need of financing, prefer trade loan financing. Because currently $\pi_m^{M^*} > \pi_m^{B^*}$, $\pi_r^{M^*} > \pi_r^{B^*}$. Under the optimal decision of trade credit, substituting the optimal equilibrium results into Formula (12) can obtain the retailer's optimal profit as $\pi_r^{M^*} = \frac{s(\alpha - bc_m)^2}{8(2bs - \beta^2)}$, the optimal profit of the retailer under bank loan financing can also be obtained as $\pi_r^{B^*} = \frac{s(\alpha - bc_m(1 + r_b))^2}{8(2bs - \beta^2)}$, $\pi_r^{M^*} - \pi_r^{B^*} = \frac{s((\alpha - bc_m) + (\alpha - bc_m(1 + r_b))bc_m r_b}{8(2bs - \beta^2)} > 0$.*

It is not difficult to understand that from Proposition 4, after retailers choose trade credit financing, sufficient funds can better carry out green marketing. The increase in market demand will be greater than the decrease in the retail price. So there will be $\pi_r^{M^*} > \pi_r^{B^*}$, the same can be obtained $\pi_m^{M^*} > \pi_m^{B^*}$. Therefore, retailers will choose trade credit financing. The further comparison found that when the interest rate of the trade credit $r_m = 0$, the optimal decision of members in the GSC will not change due to capital constraints, and the overall profit for the GSC parties will be maximized. Therefore, the trade credit financing strategy that does not charge interest is a "win-win" strategy for both parties in the GSC.

### 5. GSC Financing Strategies Under Centralized Decision-Making

*5.1. Trade Credit Financing Strategy*

Under centralized decision-making conditions, both manufacturers and retailers determine the retail product price and the level of green marketing efforts keeping the increase in total profit as a sole objective for the GSC. To maximize the GSC's overall profit, manufacturers with sufficient funds will choose to provide free loans to meet retailers' ordering needs. The total profit of the GSC is

$$\Pi^M = (p - c_m)(\alpha - bp + \beta e) - \frac{1}{2}se^2, \tag{29}$$

As can be seen, $\Pi^M$ is about the joint strict concave function of $p$ and $e$, makes its first derivative equal to 0, find the optimal equilibrium results as follows:

$$p_T^{M*} = \frac{\alpha s + (bs - \beta^2)c_m}{2bs - \beta^2}, \tag{30}$$

$$e_T^{M*} = \frac{\beta(\alpha - bc_m)}{2bs - \beta^2}, \tag{31}$$

By substituting the demand function and profit function, obtained $q_T^{M*} = \frac{bs(\alpha - bc_m)}{2bs - \beta^2}$, 

$$\tag{32}$$

$$\Pi^{M*} = \frac{s(\alpha - bc_m)^2}{2(2bs - \beta^2)}, \tag{33}$$

**Proposition 5.** *Compared with decentralized decision-making, i.e., $p^{M*} > p_T^{M*}$, $e^{M*} < e_T^{M*}$, $q^{M*} < q_T^{M*}$, $\pi_m^{M*} + \pi_r^{M*} < \Pi^{M*}$.*

By comparing with the optimal equilibrium results under decentralized decision-making, it is observed that the retail price under decentralized decision-making is higher, while the level of green marketing efforts and the order quantity of the retailer are lower, and the total profit of the GSC is also lower. An interesting finding is that $e^{M*} = \frac{1}{2}e_T^{M*}$, $q^{M*} = \frac{1}{2}q_T^{M*}$, under the decentralized decision-making of retailers, the level of green marketing efforts and the order quantity is only half of those under the centralized decision-making. Proposition 5 shows that GSC performance under decentralized decision-making is lower than centralized decision-making. Only because manufacturers and retailers are pursuing boosting their own interests under decentralized decision-making, resulting in a "double marginal effect" and low efficiency. Therefore, GSC members should strengthen cooperation to achieve the improvement of GSC efficiency jointly.

*5.2. Bank Loan Financing Strategy*

Under centralized decision-making, if bank loan financing is adopted, once the operating period is over, the retailer is required to repay the loan's principal amount and interest $\omega q(1 + r_b)$ to the bank. The total profit function of the GSC is:

$$\Pi^B = (p - c_m)q - \omega q r_b - \frac{1}{2}se^2, \tag{34}$$

The manufacturer is still in the leading position in the GSC, and the reverse recursion method is used to obtain the optimal solution. The exciting finding is that the optimal solution of the manufacturer's wholesale price changes at this time, obtained $\omega_T^{B*} = \frac{\alpha - bc_m(1 - r_b)}{2br_b}$, then compare $\omega_T^{B*}$ and $\omega^{B*}$, $\omega_T^{B*} - \omega^{B*} = \frac{(\alpha - bc_m(1 + r_b))}{2br_b(1 + r_b)} > 0$, it shows that contrasted with decentralized decision-making and centralized decision-making bank loan financing, manufacturers have increased the wholesale price in order to maximize

their own interests. While other decision variables have not changed, $p_T^{B*} = p^{B*}$, $e_T^{B*} = e_T^{B*}$, $q_T^{B*} = q^{B*}$, transformed into the profit function and found, $\Pi^{B*} = \frac{s(\alpha - bc_m(1+r_b))^2}{8(2bs - \beta^2)}$. With the increase in manufacturers' wholesale prices, retailers have to return for principal amount and interest, which has led to a decline in the total profit of the GSC.

**Proposition 6.** *Under centralized decision-making, if $r_m = r_b$, the total market demand for trade credit financing strategy is higher than the bank loan financing $\Delta q = \frac{bs(\alpha - bc_m(1-r_b))}{2(2bs - \beta^2)}$, the total profit of the GSC increases $\Delta \Pi = \frac{s\left(4(\alpha - bc_m)^2 - (\alpha - bc_m(1+r_b))^2\right)}{8(2bs - \beta^2)}$.*

Proposition 6 shows that retailers will choose trade credit financing for total profit maximization of the system GSC in centralized decision-making. Moreover, the retailer's green marketing efforts have expanded the market demand and increased the overall market demand, which is consistent with the conclusion of Proposition 3. It is not difficult to understand that under the bank's loan financing strategy, the GSC as a whole pursues the goal of profits maximization, while external financing of the GSC requires payment of loan interest to a third party, and the third party earns GSC profits which shrinks the total profit.

**Proposition 7.** *Under different financing strategies, the total profit of the GSC under centralized and decentralized decisions satisfies the following relationship:*

$$\begin{cases} \Pi^{M*} = \pi_t^{M*} + \frac{s(\alpha - bc_m)^2}{8(2bs - \beta^2)} \\ \Pi^{B*} = \pi_t^{B*} - \frac{s(\alpha - bc_m(1+r_b))^2}{4(1+r_b)(2bs - \beta^2)} \end{cases}$$

Proposition 7 shows that under the trade credit financing strategy, the total profit of the GSC with centralized decision-making is better than the decentralized decision-making. On the contrary, under the bank loan financing strategy, the decentralized decision-making's total profit is better. Based on the optimal decision knowledge, the manufacturer increased the wholesale price under the centralized decision, so the total profit decreased. Simultaneously, it can be seen from proposition 6 that the total profit of the GSC is better than that of the external financing strategy under the centralized decision-making, and the total profit of the GSC is the largest when the manufacturer and the retailer are in a cooperative alliance. In addition, the optimal decision-making of GSC under centralized decision-making can enhance consumers' environmental awareness and help develop the market. Among the two different financing strategies available to the GSC, the total profit is always most considerable under centralized decision-making. Therefore, retailers in the GSC should adopt internal financing strategy to solve the problem of capital constraints.

## 6. Numerical Analysis

In this section, numerical examples are used to analyze and illustrate the above research results. The main parameters are as follows: $\alpha = 360$, $b = 5$, $c_m = 50$, $s = 8$ and $r_n \in [0, 0.1]$. In view of different financing strategies, the influences of different interest rates, green sensitivities, and green marketing efforts on the optimal pricing decisions of GSC and the profits of member enterprises are analyzed respectively from horizontal and vertical perspectives. The specific simulation results are shown in Table 3 and Figures 1–3.

Results from Table 3 described that under decentralized decision-making. First of all, from a horizontal perspective, no matter how the green sensitivity changes, compared with no capital constraint, regardless of any financing strategy, the wholesale price of manufacturers will decline with the increase in the loan interest rate. Under the trade credit strategy, the retail price, order quantity, and green marketing level are not affected by the loan interest rate. However, under the bank loan strategy, the increase of loan interest rate

increases retailers' financing cost, weakens green marketing motivation, and reduces the order quantity. Propositions 1 and 2 are verified.

Second, from the vertical perspective, compared with no capital constraint, no matter which financing strategy, the retail price, order quantity, and green marketing level will increase with the increase of green sensitivity, the wholesale price will be unaffected. It can be further verified that under the bank loan financing strategy, the wholesale price, order quantity, and green marketing level of retailers are jointly affected by the loan interest rate and green sensitivity. Proposition 3 is also verified.

**Table 3.** Equilibrium results of different financing strategies affected by interest rates and green sensitivities under decentralized decision-making.

| Financing Strategy | | Without Capital Constraints ($i = N$) | Trade Credit ($i = M$) | | | Bank Loan ($i = B$) | | |
|---|---|---|---|---|---|---|---|---|
| $\beta \backslash r_n$ | | **0.00** | 0.03 | 0.06 | 0.09 | 0.03 | 0.06 | 0.09 |
| $\omega^{i^*}$ | 1.0 | 61.00 | 59.22 | 57.55 | 55.96 | 59.95 | 58.96 | 58.03 |
| | 3.0 | 61.00 | 59.22 | 57.55 | 55.96 | 59.95 | 58.96 | 58.03 |
| | 5.0 | 61.00 | 59.22 | 57.55 | 55.96 | 59.95 | 58.96 | 58.03 |
| $p^{i^*}$ | 1.0 | 66.57 | 66.57 | 66.57 | 66.57 | 66.94 | 67.31 | 67.68 |
| | 3.0 | 67.20 | 67.20 | 67.20 | 67.20 | 67.52 | 67.85 | 68.18 |
| | 5.0 | 69.00 | 69.00 | 69.00 | 69.00 | 69.20 | 69.41 | 69.61 |
| $e^{i^*}$ | 1.0 | 0.70 | 0.70 | 0.70 | 0.70 | 0.65 | 0.60 | 0.55 |
| | 3.0 | 2.32 | 2.32 | 2.32 | 2.32 | 2.17 | 2.01 | 1.85 |
| | 5.0 | 5.00 | 5.00 | 5.00 | 5.00 | 4.66 | 4.32 | 3.98 |
| $q^{i^*}$ | 1.0 | 27.85 | 27.85 | 27.85 | 27.85 | 25.95 | 24.05 | 22.15 |
| | 3.0 | 30.99 | 30.99 | 30.99 | 30.99 | 28.87 | 26.76 | 24.65 |
| | 5.0 | 40.00 | 40.00 | 40.00 | 40.00 | 37.27 | 34.55 | 31.82 |

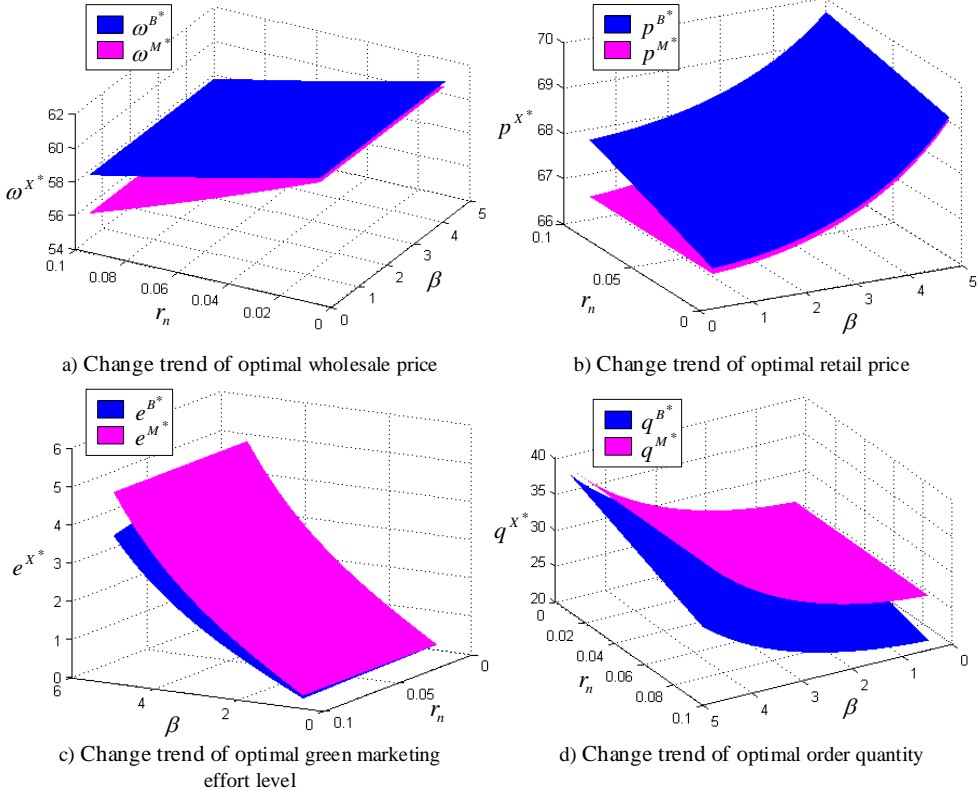

a) Change trend of optimal wholesale price

b) Change trend of optimal retail price

c) Change trend of optimal green marketing effort level

d) Change trend of optimal order quantity

**Figure 1.** Change trend of decision variables with green sensitivity and financing interest rate.

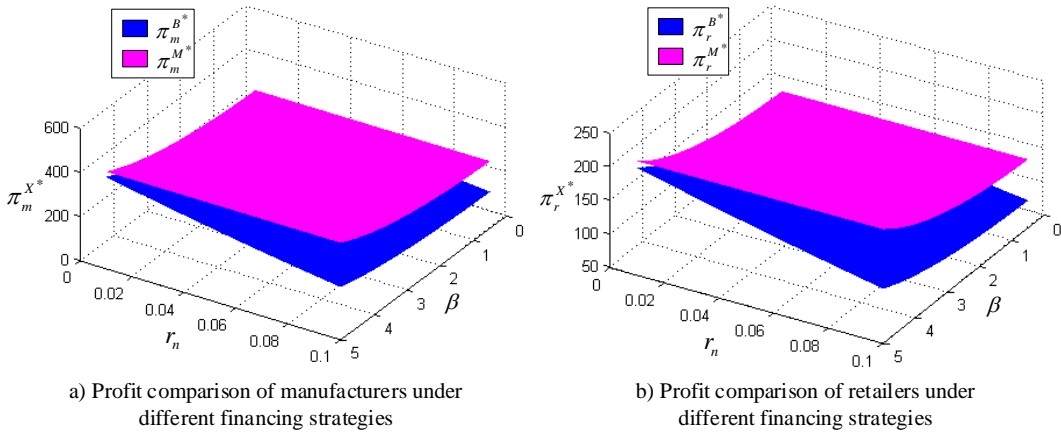

a) Profit comparison of manufacturers under different financing strategies

b) Profit comparison of retailers under different financing strategies

**Figure 2.** Profit comparison between manufacturers and retailers under decentralized Decision -making (Corollary 1).

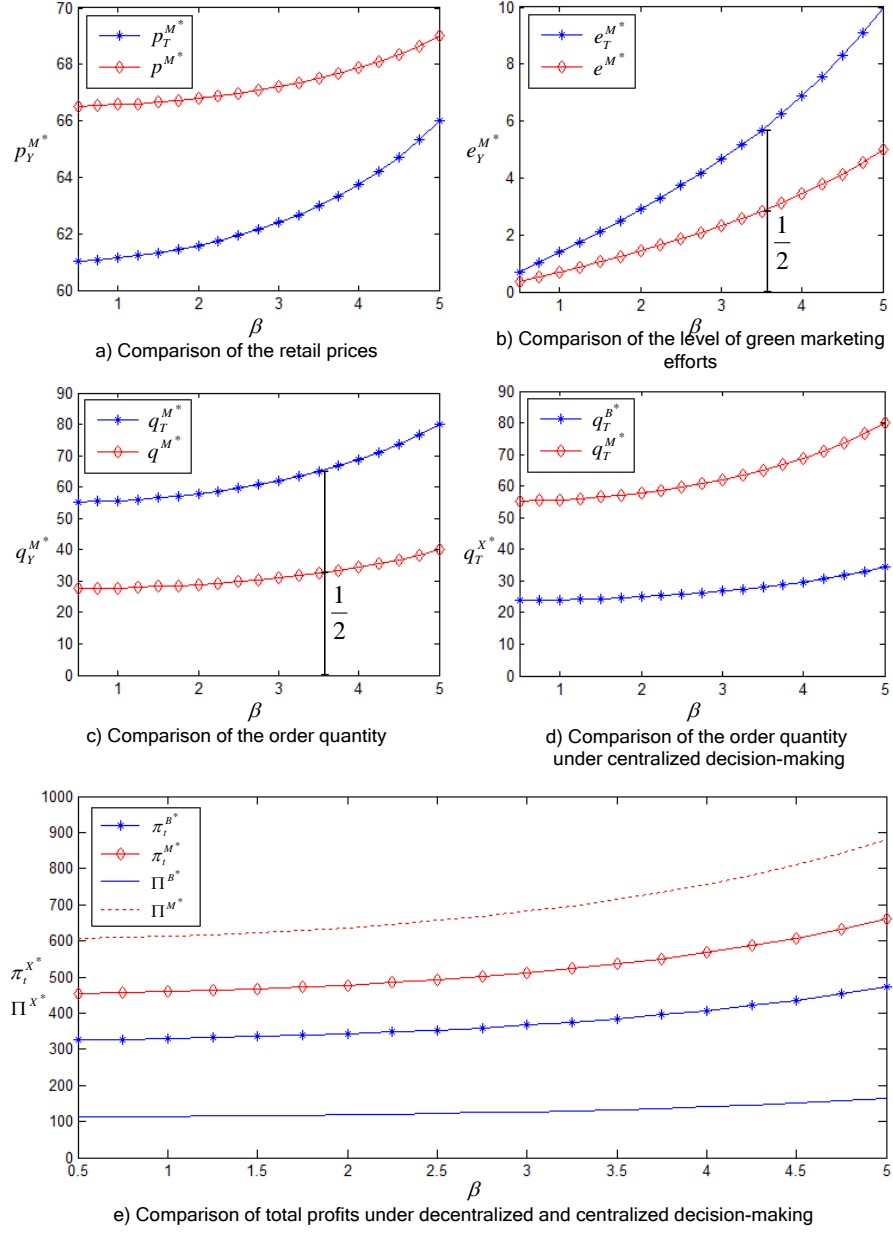

a) Comparison of the retail prices

b) Comparison of the level of green marketing efforts

c) Comparison of the order quantity

d) Comparison of the order quantity under centralized decision-making

e) Comparison of total profits under decentralized and centralized decision-making

**Figure 3.** Comparison of decision variables and total profits under decentralized and centralized decision-making ($r_m = r_b = 0.06$).

Figure 1 shows that when the trade credit interest rate is equal to the bank loan interest rate. The wholesale price and the retail price are always higher under the bank loan financing strategy. However, the level of green marketing efforts and the order quantity of retailers will increase under the trade credit financing strategy. Because under the trade credit financing strategy, manufacturers will share risks. In order to induce retailers to increase orders, wholesale prices will be reduced. The improvement of green marketing efforts will expand the market and increase the order quantity. Thus, the proposition 4 is also verified.

From Figure 2, when adopting a trade credit financing strategy, the profit of GSC member enterprises is always better than that of bank loan financing, and when the interest rate of trade credit is 0, the profit of both manufacturers and retailers reaches the highest value. Therefore, it is a win-win strategy for both sides for manufacturers to provide free trade credit to participate in retailer financing. Therefore, we accept Corollary 1.

Results from Figure 3 concluded that the loan interest rate is fixed, the value of green marketing effort level and order quantity under centralized decision-making is twice as high as that under decentralized decision-making. In addition, under the centralized decision-making, if the trade credit strategy is adopted, the order quantity will be higher, indicating that green marketing efforts have expanded the market demand and the retail price of products has decreased. The increase in the order quantity exceeds the retail price decrease, so the GSC's total profit is the largest under the centralized decision. These results verify the Proposition 5, 6, and 7.

## 7. Conclusions

This study aims to identify the level of green marketing efforts in the GSC, which will impact retailers' financing decisions under capital constraints and the profit distribution of the GSC. In this context, this paper analyzes a Stackelberg game model between a manufacturer with sufficient funds and a retailer with capital constraints in a two-level GSC (See Appendix A, for the reference model and proof of propositions established in this research). First of all, we assume that manufacturers are the dominant players in the GSC, and capital-constraints retailers maintain normal operations through financing and provide green marketing efforts to influence consumers' purchasing behavior and which may increase product sales. Second, according to the decentralized and centralized financing decision-making modes, this paper analyzes the changing trend of the wholesale price, retail price, order quantity, green marketing level with the financing interest rate, and green sensitivity under the two kinds of bank loan and trade credit financing strategies. Then, by comparing the profits of GSC participants and the overall profits of GSC under decentralized and centralized decision-making, the financing preferences of GSC participants are given. Finally, numerical examples are given to verify the outcomes of above analysis.

The results show that no matter what financing strategy is adopted, the retailer's green marketing efforts and green sensitivity are positively correlated. The relationship between green marketing efforts and financing interest rates depends on the retailer's choice of financing strategy. Conversely, if retailers choose bank loan financing, the increase in interest rates will reduce green marketing efforts. Under decentralized decision-making, manufacturers, retailers, and total profits in GSC are affected by green sensitivity and bank loan interest rates, but not by trade credit rates. It increases as green sensitivity increases and decreases as bank loan interest rates decrease. In addition, whether it is trade credit financing or bank loan financing, the optimal wholesale price has nothing to do with green sensitivity and is negatively related to financing interest rates. Moreover, the optimal retail price and the optimal order quantity increases with the increase of green sensitivity. If retailers choose bank loan financing, the increase in bank loan interest rates will increase retail prices, but the order volume will decrease. Otherwise, the retail price and order quantity will not change. Under centralized decision-making, the total profit of GSC will increase with the green sensitivity and decrease with the bank loan interest rate, but it is not affected by the trade credit interest rate.

The results show that the manufacturer's optimal wholesale price decreases with the increase in the loan interest rate regardless of the retailer's financing method under the decentralized decision-making. If the retailer chooses trade credit financing, the optimal retail price, green marketing efforts level, order quantity, and these decision variables are irrelevant to the financing interest rate and equal to the value without capital constraint. If the retailer chooses bank loan financing, with the increase in the loan interest rate, the optimal retail price will increase, while green marketing efforts and order quantity will decrease. In centralized decision-making, if the retailer adopts trade credit financing, the optimal retail price will be lower than that in decentralized decision-making. The optimal green marketing efforts level and the optimal order quantity will be twice as high as those in decentralized decision-making. The total profit of the GSC will be higher than that in the decentralized decision-making. If the retailer adopts bank loan financing, the manufacturer's optimal wholesale price will be higher than the trade credit financing, while the retailer's optimal order quantity and the total profit of the GSC will be lower than that of the decentralized decision.

This paper has the following implications: This research attempts to enhance the green supply chain's existing knowledge, contemplating the more accurate backdrop of the optimal financing strategy. This research is the initial step of stimulus design in the literature of capital-constraint supply chain and green marketing effort. Besides the contributions to the literature, this research also derives some managerial insights. First, under the decentralized decision, if the interest rate of trade credit is equal to that of bank loans, both manufacturers and retailers tend to choose trade credit financing to improve their profits. In particular, if manufacturers provide interest-free trade credit, the profits of members and the whole GSC can be maximized at the same time, which is a "win-win" strategy for both sides of the GSC. Second, under centralized decision-making, retailers still prefer trade credit financing. By comparing the profits of the enterprises and the total profit of the GSC system under the decentralized and centralized decision-making, the centralized decision-making to overcome the influence of the "double marginal effect". Third, with the use of the internal GSC financing strategy, manufacturers and retailers in the alliance can make GSC the largest gross profit. Therefore, no matter in a decentralized or centralized decision-making model, retailers in the GSC should adopt trade credit, that is, GSC internal financing strategy, to solve their capital constraint problem.

**Author Contributions:** Conceptualization, X.Z.; methodology, X.Z.; software, X.Z. and Y.D.; validation, X.Z., F.S. and Y.D.; formal analysis, X.Z.; resources, G.X.; writing—original draft preparation, X.Z. and F.S.; writing—review and editing, F.S. and Y.D.; supervision, G.X.; funding acquisition, G.X. All authors have read and agreed to the published version of the manuscript.

**Funding:** The National Natural Science Foundation of China (72001059); The Philosophy and Social Science Fund Project of Heilongjiang Province (19JYD189); The Provincial Undergraduate Universities Fundamental Scientific Research Business Expense Project of Heilongjiang Province (2019-KYYWF-004).

**Institutional Review Board Statement:** Not applicable.

**Informed Consent Statement:** Not applicable.

**Data Availability Statement:** Not applicable.

**Conflicts of Interest:** The authors declare no conflict of interest.

## Appendix A. Optimal Equilibrium Solution of the GSC without Capital Constraint

According to the order of the two-stage GSC Stackelberg game model, manufacturer leader and retailer follower adopt the reverse recursive method, which is easy to verify,

$$\frac{\partial^2 \pi_r^N(p,e)}{\partial p^2} = -2b < 0, \quad \frac{\partial^2 \pi_r^N(p,e)}{\partial e^2} = -s < 0,$$

We can get that the Hessian Matrix of $\pi_r^N(p,e)$ as follows:

$$H_{(p,e)} = \begin{bmatrix} -2b & \beta \\ \beta & -s \end{bmatrix}.$$

When $D = 2bs - \beta^2 > 0$, $\pi_r^N(p,e)$ is all about the joint strictly concave function of $p$ and $e$, equating $\frac{\partial \pi_r^N(p,e)}{\partial p} = 0$ and $\frac{\partial \pi_r^N(p,e)}{\partial e} = 0$, we obtain the optimal response function of the retailer:

$$p^N(\omega) = \frac{\alpha s + (bs - \beta^2)\omega}{2bs - \beta^2}. e^N(\omega) = \frac{\beta(\alpha - b\omega)}{2bs - \beta^2}.$$

where $p^N(\omega)$ and $e^N(\omega)$ substitute it into the manufacturer's profit function, similarly, it is easy to verify $\pi_m^N(\omega)$ whereas $\omega$ is a concave function, the first derivative is zero, and the optimal wholesale price of the manufacturer when the retailer has sufficient funds and no financing needs is:

$$\omega^{N*} = \frac{\alpha + bc_m}{2b}.$$

whereas $\omega^{N*}$ substitute $p^N(\omega)$, $e^N(\omega)$ and the demand function, the retail price, the level of green marketing efforts, and the order quantity of the retailer can be obtained respectively:

$$p^{N*} = \frac{2\alpha bs + (bs - \beta^2)(\alpha + bc_m)}{2b(2bs - \beta^2)} e^{N*} = \frac{\beta(\alpha - bc_m)}{2(2bs - \beta^2)} q^{N*} = \frac{bs(\alpha - bc_m)}{2(2bs - \beta^2)}$$

**Proof of Proposition 1.** The optimal solution process is similar to the solution process without capital constraint and is not listed in detail.

$$\omega^{M*} - \omega^{N*} = -\frac{(\alpha + bc_m)r_m}{2b(1 + r_m)} < 0, \text{ so} \omega^{M*} < \omega^{N*}.$$

$\square$

**Proof of Proposition 2.** $\frac{\partial \omega^{B*}}{\partial r_b} = \frac{-\alpha}{2b(1+r_b)^2} < 0$, $\frac{\partial p^{B*}}{\partial r_b} = \frac{(bs-\beta^2)c_m}{2(2bs-\beta^2)} > 0$, $\frac{\partial e^{B*}}{\partial r_b} = \frac{-b\beta c_m}{2(2bs-\beta^2)} < 0$, $\frac{\partial q^{B*}}{\partial r_b} = \frac{-b^2 s c_m}{2(2bs-\beta^2)} < 0$. Proposition 2 is proved. $\square$

**Proof of Proposition 3.** When $X = M$, then $\frac{\partial \omega^{M*}}{\partial \beta} = 0$, $\frac{\partial p^{M*}}{\partial \beta} = \frac{s\beta(\alpha-bc_m)}{2bs-\beta^2} > 0$, $\frac{\partial e^{M*}}{\partial \beta} = \frac{(\alpha-bc_m)(2bs+\beta^2)}{2(2bs-\beta^2)^2} > 0$, $\frac{\partial q^{M*}}{\partial \beta} = \frac{bs\beta(\alpha-bc_m)}{(2bs-\beta^2)^2} > 0$.

When $X = B$ then the proof process is similar, it is not stated anymore. $\square$

**Proof of Proposition 4.** When $r_m = r_b$ then, $\omega^{B*} - \omega^{M*} = \frac{c_m r_b}{2(1+r_b)} \geq 0$, $p^{B*} - p^{M*} = \frac{(bs-\beta^2)c_m r_b}{2(2bs-\beta^2)} \geq 0$, $e^{B*} - e^{M*} = -\frac{b\beta c_m r_b}{2(2bs-\beta^2)} \leq 0$, $q^{B*} - q^{M*} = -\frac{b^2 s c_m r_b}{2(2bs-\beta^2)} \leq 0$. Proposition 4 is proved. $\square$

**Proof of Proposition 5.** $p^{M*} - p_T^{M*} = \frac{(bs-\beta^2)(\alpha-bc_m)}{2b(2bs-\beta^2)} > 0$, $e^{M*} - e_T^{M*} = \frac{-\beta(\alpha-bc_m)}{2b(2bs-\beta^2)} < 0$, $q^{M*} - q_T^{M*} = \frac{-bs(\alpha-bc_m)}{2b(2bs-\beta^2)} < 0$, as per the sum of Formulas (11) and (12), we obtained $\pi_t^{M*} - \Pi^{M*} = -\frac{s(\alpha-bc_m)^2}{8(2bs-\beta^2)} < 0$, then $\pi_t^{M*} - \Pi^{M*} = -\frac{s(\alpha-bc_m)^2}{8(2bs-\beta^2)} < 0$. Proposition 5 is proved. $\square$

**Proof of Proposition 6.** $\Delta q = q_T^{M*} - q_T^{B*} = \frac{bs(\alpha - bc_m(1-r_b))}{2(2bs-\beta^2)}$, $\Delta \Pi = \Pi^{M*} - \Pi^{B*} = \frac{s(4(\alpha-bc_m)^2 - (\alpha-bc_m(1+r_b))^2)}{8(2bs-\beta^2)}$. $\square$

**Proof of Proposition 7.** According to the sum of Equations (11) and (12), $\pi_t^{M*} = \frac{3s(\alpha - bc_m)^2}{8(2bs - \beta^2)}$, while as per Formula (29) $\Pi^{M*} = \frac{s(\alpha - bc_m)^2}{2(2bs - \beta^2)}$, relatively available $\Pi^{M*} = \pi_t^{M*} + \frac{s(\alpha - bc_m)^2}{8(2bs - \beta^2)}$; similarly $\Pi^{B*} = \pi_t^{B*} - \frac{s(\alpha - bc_m(1 + r_b))^2}{4(1 + r_b)(2bs - \beta^2)}$. □

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
