# Peer review of "Optimal Financing Strategy in a Capital-Constrained Supply Chain with Retailer Green Marketing Efforts"

_sustainability, doi:10.3390/su13031357_

Round 1
Reviewer 1 Report
The paper is well written based on the relevant and up-to date literature review., respecting scientific standards of writing articles.
Methodology used in the paper is suitable, the sources of information and the measurement tools are adequately defined.
Gained results are described followed by the discussion part emphasizing all results.
The conclusion part contains also limitations and future research posibilities.
Author Response
Thank you very much for your quality time and efforts in reading and accepting our manuscript.
Reviewer 2 Report
The paper is well written and it focuses on an interesting topic. Minor changes are related to the need of specifying into the introduction which is the gap that the paper aims to overcome. Secondly, some formula in the text of the paragraph on simulation present the "error" and are not present. Thirdly, while overall the manuscript is largely described, the section conclusions (that includes also the discussions) is more synthetic and could be a little bit extended in order to provide evidences and implications for theory.
Author Response
Reviewer 2 comments. The paper is well written and it focuses on an interesting topic. Minor changes are related to the need of specifying into the introduction which is the gap that the paper aims to overcome. Secondly, some formula in the text of the paragraph on simulation present the "error" and are not present. Thirdly, while overall the manuscript is largely described, the section conclusions (that includes also the discussions) is more synthetic and could be a little bit extended in order to provide evidences and implications for theory Response: Thank you very much for your kind consideration. We have made all the changes as suggested in our manuscript. Research gap is more clearly defined in the introduction. The formula in the text are double checked and corrected throughout the manuscript. Finally, we have revised and extended our conclusion section accordingly.Reviewer 3 Report
To whom may concern,
Please find my observations:
- There are some paragraphs of the paper that do not seem to be connected - lack of cohesion in some parts, specifically in the Introduction. E.g. abruptly move from green products to green marketing - the theoretical parts of the paper read more like a list of concept, not a text with clear structure - Needs improvement
- You miss identifying some key concepts in the paper. E.g. What is a green product? What is the Stackelberg model?
- Missing references along with the paper - e.g. Walmart example
- I believe you haven't answered the 2nd research question. It would be good if you could refer to the RQs in the conclusion section
- Missing clarity on the research methodology - why did you use the Stackelberg model? What are the other models available?
- References style changes along with the paper. E.g. With the increasing awareness of environmental protection, consumers are willing 153 to choose green products as an effective way to protect the environment [2,33]. The study 154 of Ko et al. (2013) posits green marketing in products and sales from the perspective of 155 consumers.
- Several error messages: Error! Reference source not found
To summarise, I believe that the paper touches an important topic, but it lacks in academic depth. The paper needs clear improvements in the Introduction, Literature review sections and conclusion. The empirical element seems fine, but, again, it lacks in demonstrating the impact (and referring to the RQs/ introduction).
Author Response
Reviewer 3 comments and responses
Please find my observations:
- There are some paragraphs of the paper that do not seem to be connected - lack of cohesion in some parts, specifically in the Introduction. E.g. abruptly move from green products to green marketing - the theoretical parts of the paper read more like a list of concept, not a text with clear structure - Needs improvement
Response:
We have further improved the structure of the introduction and manuscript as suggested.
- You miss identifying some key concepts in the paper. E.g. What is a green product? What is the Stackelberg model?
Response:
In the revised manuscript, we have incorporated the definitions of key concepts. Green production in the first paragraph of the introduction and the Stackelberg model in the last paragraph of the introduction.
- Missing references along with the paper - e.g. Walmart example
Response:
We have added the missing reference as you highlighted.
- I believe you haven't answered the 2nd research question. It would be good if you could refer to the RQs in the conclusion section
Response:
We have incorporated the conclusion about the 2nd research questions in section 7. Conclusion, para 2.
- Missing clarity on the research methodology - why did you use the Stackelberg model? What are the other models available?
Response:
We have further clarified the research methodology.
The Stackelberg model can solve the perfect Nash equilibrium or equilibrium of the subgame, that is, given the strategies of other players, the most advantageous strategy for each player requires each player to be in the Nash equilibrium state in each subgame. In this research setting, because the manufacturer has sufficient funds and sufficient capacity to produce green products, it has an obvious strong position in the green supply chain, while the retailer has insufficient funds and is in a relatively weak position, so the manufacturer has more priority.
Moreover, the Stackelberg game model has been used by several scholars in similar research settings, e.g. (Ferrara et al., 2017; Savaskan et al., 2004; Wu et al., 2020; Yan et al., 2016; Zhang and Wang, 2019). So the Stackelberg game model is the best approach in this situation.
- References style changes along with the paper. E.g. With the increasing awareness of environmental protection, consumers are willing 153 to choose green products as an effective way to protect the environment [2,33]. The study 154 of Ko et al. (2013)posits green marketing in products and sales from the perspective of 155 consumers.
Response:
We have revised all the references style as per the journal guidelines.
- Several error messages: Error! Reference source not found
Response:
We have double checked for such errors.
To summarise, I believe that the paper touches an important topic, but it lacks in academic depth. The paper needs clear improvements in the Introduction, Literature review sections, and conclusion. The empirical element seems fine, but, again, it lacks in demonstrating the impact (and referring to the RQs/ introduction).
Response:
Thank you very much for your consideration. We have considered your valuable suggestions and revised our manuscript accordingly. We hope it will get the chance of publication.
Round 2
Reviewer 3 Report
There is a clear improvement in the paper. In particular, the conclusion is the section that has been most improved.